# Novel *SCN5A* p.W697X Nonsense Mutation Segregation in a Family with Brugada Syndrome

**DOI:** 10.3390/ijms20194920

**Published:** 2019-10-04

**Authors:** Emanuele Micaglio, Michelle M. Monasky, Nicoletta Resta, Rosanna Bagnulo, Giuseppe Ciconte, Luigi Gianelli, Emanuela T. Locati, Gabriele Vicedomini, Valeria Borrelli, Andrea Ghiroldi, Luigi Anastasia, Sara Benedetti, Chiara Di Resta, Maurizio Ferrari, Carlo Pappone

**Affiliations:** 1Arrhythmology Department, IRCCS Policlinico San Donato, San Donato Milanese, 20097 Milan, ItalyMichelle.monasky@grupposandonato.it (M.M.M.); g.ciconte@gmail.com (G.C.); giannelli.luigi@gmail.com (L.G.); EmanuelaTeresina.Locati@grupposandonato.it (E.T.L.); Gabriele.Vicedomini@grupposandonato.it (G.V.); valiborrelli91@gmail.com (V.B.); 2Medical Genetics Unit, Department of Biomedical Sciences and Human Oncology, “Aldo Moro” University of Bari, Policlinico Hospital, 70121 Bari, Italy; nicoletta.resta@uniba.it (N.R.); rosanna.bagnulo@uniba.it (R.B.); 3Stem Cells for Tissue Engineering Laboratory, IRCCS Policlinico San Donato, San Donato Milanese, 20097 Milan, Italy; andrea.ghiroldi@gmail.com (A.G.); luigi.anastasia@unimi.it (L.A.); 4Department of Biomedical Sciences for Health, University of Milan, 20122 Milan, Italy; 5Laboratory of Clinical Molecular Biology and Cytogenetics, Division of Genetics and Cellular Biology, IRCCS San Raffaele Hospital, 20132 Milan, Italy; Benedetti.sara@hsr.it (S.B.); ferrari.maurizio@hsr.it (M.F.); 6Genomic Unit for the Diagnosis of Human Pathologies, Division of Genetics and Cellular Biology, IRCCS San Raffaele Hospital, 20132 Milan, Italy; diresta.chiara@hsr.it; 7Vita-Salute San Raffaele University, 20132 Milan, Italy

**Keywords:** Brugada syndrome, sudden cardiac death, genetic testing, mutation, *SCN5A*, sodium channel, arrhythmia, channelopathy, family, point-nonsense mutation

## Abstract

Brugada syndrome (BrS) is marked by an elevated ST-segment elevation and increased risk of sudden cardiac death. Variants in the *SCN5A* gene are considered to be molecular confirmation of the syndrome in about one third of cases, while the genetics remain a mystery in about half of the cases, with the remaining cases being attributed to variants in any of a number of genes. Before research models can be developed, it is imperative to understand the genetics in patients. Even data from humans is complicated, since variants in the most common gene in BrS, *SCN5A,* are associated with a number of pathologies, or could even be considered benign, depending on the variant. Here, we provide crucial human data on a novel NM_198056.2:c.2091G>A (p.Trp697X) point-nonsense heterozygous variant in the *SCN5A* gene, as well as its segregation with BrS. The results herein suggest a pathogenic effect of this variant. These results could be used as a stepping stone for functional studies to better understand the molecular effects of this variant in BrS.

## 1. Introduction

The Brugada syndrome (BrS) is diagnosed by the presence of a coved-type ST-segment elevation in the right precordial leads on the electrocardiogram (ECG), which may occur either spontaneously or after administration of a sodium channel blocking agent, such as ajmaline [1]. The syndrome is associated with an increased risk of sudden cardiac death (SCD) [1,2] and is generally considered to be inherited as an autosomal dominant disease with incomplete penetrance [3,4,5]. However, there are a few articles reporting alternative inheritance mechanisms [6,7], such as recessive and X linked inheritance. Additionally, a recent study shed light on the possible role of mitochondrial mutations [8]. The genetics of BrS are not well understood, with about half of cases lacking molecular confirmation, and possible new genes are an active area of research [9,10,11,12,13]. About 15–30% of cases are thought to be caused by variants in the gene *SCN5A*, which is the most commonly associated gene with BrS [14] and is associated with a loss of function of the voltage-gated sodium channel subunit (Na_v_1.5) [13,15,16]. However, variants in *SCN5A* are associated with many pathologies, including atrial standstill type 1, atrial fibrillation, left ventricular noncompaction, dilated cardiomyopathy, long QT syndrome type 3, sick sinus syndrome type 2, idiopathic ventricular fibrillation, and heart block type 1A [2,17,18], making genotype-phenotype predictions difficult without sufficient clinical data.

Novel mutations in the *SCN5A* gene that are likely responsible for BrS have been recently of interest [10,19,20,21,22,23]. Given that genetic variants are not detected in the majority of BrS patients [24], and even if a variant is detected, the significance of many variants is uncertain; understanding better the genetics of BrS could be useful for risk stratification. However, research studies are limited by gaps in the clinical data.

Understanding the genetics better is important to identify patients who may not even be aware that they have BrS. SCD is sometimes the first indication that someone may have been affected by BrS. Thus, the true prevalence is uncertain, and difficulties in making a diagnosis are compounded by a wide degree of clinical variability, even within the same family [2], and the invasive nature, risks [25], and costs of ajmaline testing, sometimes deter patients. There have even been reports of false-positive ajmaline testing, making genetic diagnosis instrumental in determining whether these patients actually have the disease [26].

In this study, the variant NM_198056.2:c.2091G>A (p.Trp697X) in the *SCN5A* gene is characterized for the first time in a family with BrS, providing crucial human data that could be used as a stepping stone to advance diagnostic capabilities.

## 2. Results

### 2.1. Case Presentation

The study was conducted in accordance with the Declaration of Helsinki, and written informed consent of human subjects was obtained for their participation in the study and for publication. The procedures employed were reviewed and approved by the local Ethics Committee (approver number: M-EC-006/A, 29 August 2019). The proband was a 30-year-old male of Italian descent with a history of palpitations and syncope since puberty. He received a diagnosis of BrS elsewhere due to a spontaneous type 1 BrS ECG pattern. His brother had also been diagnosed with BrS elsewhere. Thus, he had an implantable cardioverter defibrillator (ICD) implanted. Echocardiographic examination demonstrated normal morphological and functional parameters. When he came to our attention for a second opinion and radiofrequency ablation of the arrhythmogenic substrate, genetic testing revealed the variant NM_198056.2:c.2091G>A (p.Trp697X) in the *SCN5A* gene (Leiden Open Variation Database: https://databases.lovd.nl/shared/variants/0000478717#00018523) (Figure 1). A spontaneous type 1 ECG pattern was observed, and he underwent ablation of the arrhythmogenic substrate (AS) (Figure 2). 

### 2.2. Assessment of Family Members

The family pedigree can be seen in Figure 3. The proband’s 58-year-old mother had a history of hypothyroidism, arterial hypertension, and obesity. She was found to harbor the same NM_198056.2:c.2091G>A (p.Trp697X) variant in the *SCN5A* gene as her children. At this point, she came to our attention and underwent both ajmaline challenge and electrophysiological study (EPS), which were both positive (Figure 4A). An ICD was subsequently implanted. Her brother was subjected to genetic testing and a flecainide challenge elsewhere, both of which were negative.

The proband’s 33-year-old sister underwent genetic counseling elsewhere, and was found to harbor the same NM_198056.2:c.2091G>A (p.Trp697X) variant in the *SCN5A* gene that had been found in her brothers. When she came to our attention, she underwent an ajmaline challenge, which confirmed a BrS diagnosis. An EPS was performed, which was also positive, and an ICD was implanted (Figure 4B).

### 2.3. In Silico Predictions

The VarSome [23,24,27] genetic database was used to predict the significance of the *SCN5A* variant. The genomic evolutionary rate profiling (GERP) value was noted, which is defined by VarSome as “a conservation score calculated by quantifying substitution deficits across multiple alignments of orthologues using the genomes of 35 mammals. It ranges from –12.3 to 6.17, with 6.17 being the most conserved.” [28]. The GERP score for the NM_198056.2:c.2091G>A (p.Trp697X) variant in the *SCN5A* gene is 4.9, indicating that this variant has occurred in a highly conserved area of the genome, suggesting that variants in this region could be pathogenic. In fact, this variant is classified as pathogenic in VarSome, with a “disease causing automatic” result from MutationTaster, a “damaging” result from functional analysis Through hidden Markov models (FATHMM) - Multiple Kernel Learning (MKL), and a deep artificial neural network (DANN) score of 0.9969. DANN scores range from 0 to 1, with 1 given to the variants predicted to be the most damaging [29]. The likelihood ratio test (LRT) predicts deleterious variants through identification of highly conserved amino acid regions using a comparative genomics data set of 32 vertebrate species, and ranges from 0 to 1 [27]. The LRT score for the NM_198056.2:c.2091G>A (p.Trp697X) variant in the *SCN5A* gene is 0, with a prediction of “deleterious” [27]. A multicenter study by Kapplinger et al. described a BrS-related mutation in the *SCN5A* gene located on the nucleotide immediately after the nucleotide of the variant described in the current study (c.2092G>T, p.E698X*), lending further evidence to the pathogenicity of the c.2091G>A variant [14].

The c.2091G>A variant was classified as pathogenic according to ACMG criteria [27]:

PVS1: Null variant (nonsense) affecting gene *SCN5A*, which is a known mechanism of disease (385 pathogenic variants out of 749 classified variants = 51.40%, which is greater than threshold = 5.0%), associated with Brugada syndrome, atrial fibrillation, long QT syndrome type 3, idiopathic ventricular fibrillation, progressive heart block, nonprogressive heart block, sick sinus syndrome, and dilated cardiomyopathy.

PM2: Variant not found in GnomAD exomes (good GnomAD exomes coverage = 72.2). Variant not found in GnomAD genomes (good GnomAD genomes coverage = 34.4).

PP3: Pathogenic computational verdict based on five pathogenic predictions from DANN, GERP, LRT, MutationTaster, and FATHMM-MKL (vs no benign predictions).

## 3. Discussion

In the present study, we report for the first time the NM_198056.2:c.2091G>A (p.Trp697X) heterozygous variant in the *SCN5A* gene and its segregation with BrS. Collectively, the results herein support a likely pathogenic effect of this variant and provide a stepping stone to advance our diagnostic capabilities in patients with this variant.

The clinical picture in these family members is important to emphasize. The proband exhibits a spontaneous type 1 BrS ECG pattern, which many studies have reported to possibly increase the risk of arrhythmic events, and he has experienced palpitations and syncope since puberty. The presence of both syncope and a spontaneous type 1 ECG pattern has been described as a strong indicator of poor prognosis during follow-up (6% to 19% of patients will have an arrhythmic event within 24 to 39 months of follow-up) [9,30]. The area of the arrhythmogenic substrate [31,32], measured before epicardial radiofrequency ablation, measured 9.7 cm^2^, and is denoted by the “marked area” in Figure 2. A minimal area of 4 cm^2^ has been generally associated with inducibility for ventricular tachycardia/fibrillation during the electrophysiological study [33], which is associated with future ventricular arrhythmia risk [34]. Thus, although the proband did not perform an electrophysiological study, the area of the arrhythmogenic substrate would suggest a concerning prognosis. The proband’s mother and sister were both inducible for the development of ventricular tachycardia/fibrillation during the electrophysiological study. Thus, the inducibility and symptoms together, seen in various family members with this genetic variant, suggest a likely pathogenic role for the c.2091G>A variant.

Most of the research into the molecular pathology of BrS has focused on the alpha subunit of the Na_V_1.5 protein, codified by the *SCN5A* gene and generally believed to exhibit reduced function in BrS. Reduced expression in the sarcolemma [35,36,37], production of non-functional channels [38], and alterations in gating properties [39,40] have been described as mechanisms that lead to the reduced function of this channel. The various pathologies that result from *SCN5A* variants may be explained, at least in part, by the various types of mutations that may occur, namely missense, nonsense, splicing, insertion/deletion, and frameshift [2,14]. The variant c.2091G>A described in the present study is a nonsense mutation that results in a premature stop codon and a truncated and incomplete protein product, a consequence that is predicted to be likely pathogenic. Several other studies have reported pathogenic effects of premature stop codons in the *SCN5A* gene, leading to reduced protein expression, a non-functional protein that was confined in the cytosol rather than reaching the plasma membrane, and a complete loss of current [35,36,41]. 

The pathogenicity of the c.2091G>A variant is supported by the fact that it is found in a gene that is known to be responsible for various pathologies (Brugada syndrome, familial atrial fibrillation, long QT syndrome type 3, idiopathic ventricular fibrillation, complete heart block, sick sinus syndrome, dilated cardiomyopathy), its extreme rarity in the general population (not found in GnomAD exomes or genomes, despite good coverage), and the unanimous pathogenic in silico predictions (from DANN, GERP, LRT, MutationTaster, and FATHMM-MKL). These results together strongly support a pathogenic role for this variant.

## 4. Concluding Remarks

The novel heterozygous variant NM_198056.2:c.2091G>A (p.Trp697X) in the *SCN5A* gene segregates with BrS in the family presented, providing crucial human data relevant to understanding the pathology of BrS for patients with this variant. The results herein suggest a likely pathogenic effect of this variant and could be used as a stepping stone for functional studies to better understand the molecular pathways involved.

## Figures and Tables

**Figure 1 ijms-20-04920-f001:**
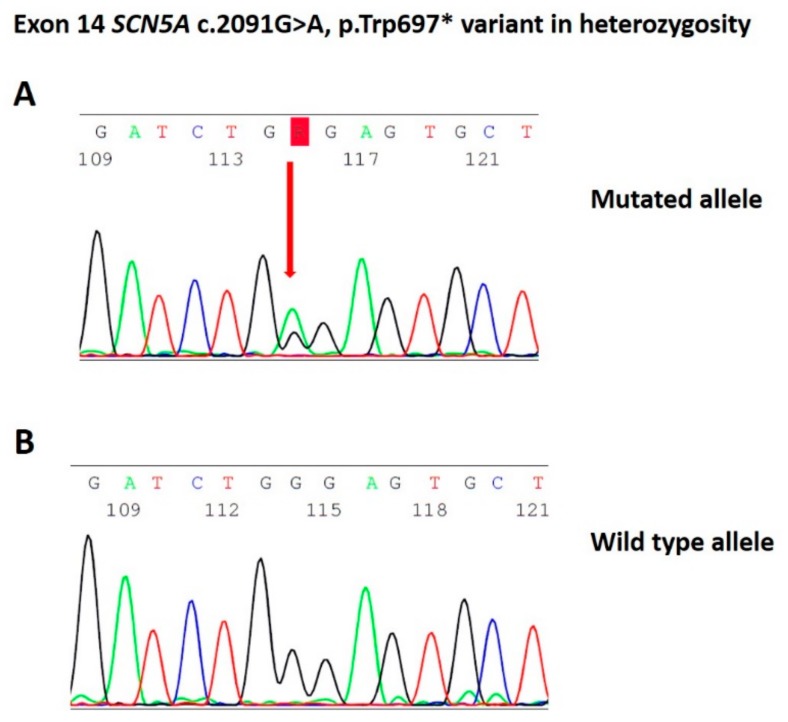
(**A**) Identification of the heterozygous c.2091G>A mutation by Sanger sequencing. (**B**) Negative control from proband’s family identified by Sanger sequencing.

**Figure 2 ijms-20-04920-f002:**
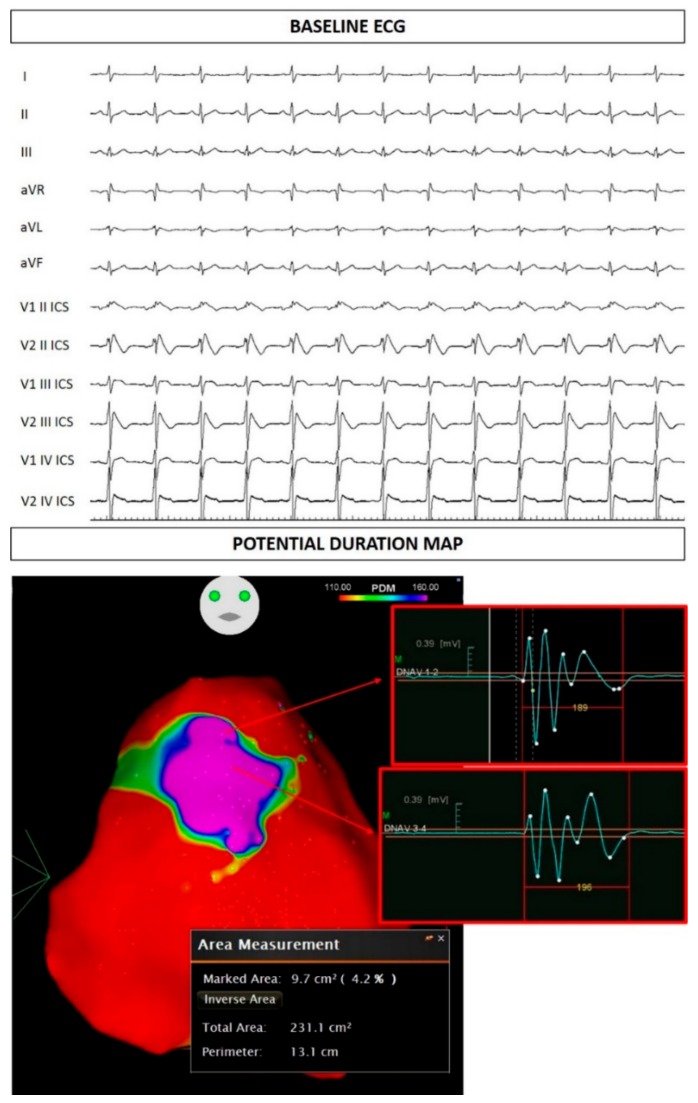
Proband electrocardiogram (ECG) at baseline. Potential duration map demonstrating the epicardial arrhythmogenic substrate.

**Figure 3 ijms-20-04920-f003:**
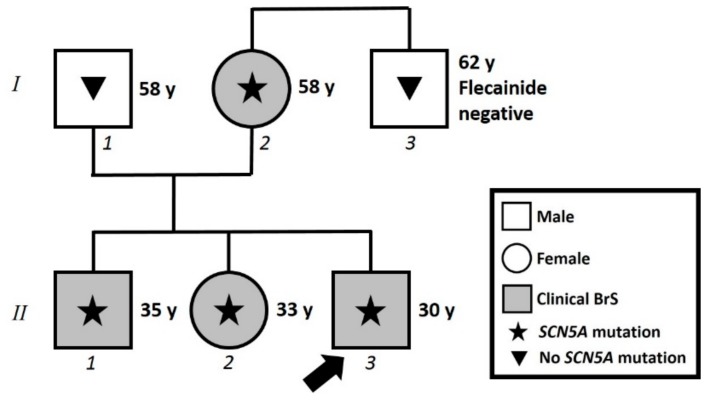
Family pedigree. Proband identified with arrow. Square: male; Circle: female; Shaded: clinically affected by Brugada syndrome; Star: molecularly confirmed *SCN5A* mutation: Triangle: genetically tested and negative for *SCN5A* mutation. y = years old at diagnosis.

**Figure 4 ijms-20-04920-f004:**
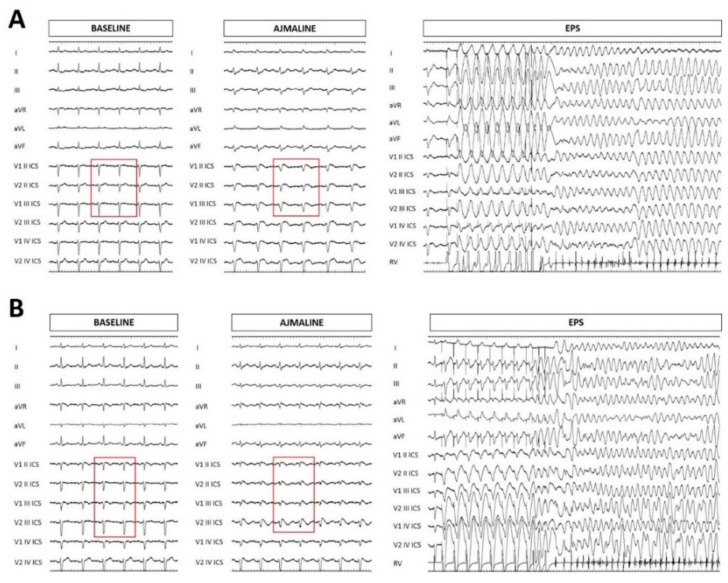
Electrocardiogram at baseline, after ajmaline administration, and ventricular tachycardia/ventricular fibrillation inducibility during electrophysiological study (EPS) for the proband’s mother (**A**) and sister (**B**). The BrS pattern is not seen at baseline, but is seen after ajmaline administration (red rectangles).

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
