# Peer review of "Novel SCN5A p.W697X Nonsense Mutation Segregation in a Family with Brugada Syndrome"

_ijms, 2019, doi:10.3390/ijms20194920_

Round 1

Reviewer 1 Report

This case report describes a novel mutation in the SCN5A gene in a family with Brugada syndrome.

In silico prediction is used to assess whether this mutation may contribute to the disease.

This study may help understanding the pathophysiology and molecular mechanisms of this disease.

There is however several points to consider.

The Introduction should present a state-of-the-art of Brugada syndrome genetics in regard with other non-genetic causes of the disease, if any. What is the current knowledge about the disease pathophysiology? Part of the discussion should be moved to the Introduction.

Is the variant NM_198056.2:c.2091G>A (p.Trp697X) the only variant identified in the SCN5A gene in this family? Has the SCN5A gene been fully sequenced?

The in silico prediction method should be more detailed. What is the degree of certainty of such analyses? Since the described genetic variant is a nonsense mutation that results in a premature stop codon and a truncated and incomplete protein product, the utility of these analyses to predict the gene variant pathogenicity can be questioned.

Could this truncated protein be active? Could it be expressed as an integral membrane protein? Could the expression of this truncated protein affect the function of the non mutated channel? Structure-function protein modeling should have been done or at least some mechanistic clues should be provided in the discussion.

Author Response

Point 1: The Introduction should present a state-of-the-art of Brugada syndrome genetics in regard with other non-genetic causes of the disease, if any. What is the current knowledge about the disease pathophysiology? Part of the discussion should be moved to the Introduction.

Response 1: Brugada Syndrome is believed to be inherited as an autosomal dominant disease with incomplete penetrance. We have now included in the introduction, “However, there are a few articles reporting alternative inheritance mechanisms (PMID: 30142439, 21493962), such as recessive and X linked inheritance. Additionally, a recent study shed light on a possible role of mitochondrial mutations (PMID: 28980288).” To date, non-genetic causes of Brugada syndrome have never been described. Currently, the research in the field is dedicated to clarify better the genetics, to understand new causative genes, and to assess which variants in those genes are actually pathogenic. Other areas of study involve understanding the many phenotypes that might be caused by particular pathogenic variants (i.e. variants in SCN5A can result in a variety of phenotypes, as explained in the introduction).

Point 2: Is the variant NM_198056.2:c.2091G>A (p.Trp697X) the only variant identified in the SCN5A gene in this family? Has the SCN5A gene been fully sequenced?

Response 2: The answers are yes to both questions: This is the only variant in the SCN5A gene identified in this family, and the gene has been fully sequenced.

Point 3: The in silico prediction method should be more detailed. What is the degree of certainty of such analyses? Since the described genetic variant is a nonsense mutation that results in a premature stop codon and a truncated and incomplete protein product, the utility of these analyses to predict the gene variant pathogenicity can be questioned.

Response 3: We have now added the following discussion to the in silico prediction section: “The likelihood ratio test (LRT) predicts deleterious variants through identification of highly conserved amino acid regions using a comparative genomics data set of 32 vertebrate species, and ranges from 0 to 1 (Varsome). The LRT score for the NM_198056.2:c.2091G>A (p.Trp697X) variant in the SCN5A gene is 0, with a prediction of “deleterious” (Varsome). A multicenter study by Kapplinger and colleagues described a BrS-related mutation in the SCN5A gene located on the nucleotide immediately after the nucleotide of the variant described in the current study (c.2092G>T, p.E698X*), lending further evidence to the pathogenicity of the c.2091G>A variant (Kapplinger JD, Tester DJ, Alders M, Benito B, Berthet M, Brugada J, Brugada P, Fressart V, Guerchicoff A, Harris-Kerr C, Kamakura S, Kyndt F, Koopmann TT, Miyamoto Y, Pfeiffer R, Pollevick GD, Probst V, Zumhagen S, Vatta M, Towbin JA, Shimizu W, Schulze-Bahr E, Antzelevitch C, Salisbury BA, Guicheney P, Wilde AA, Brugada R, Schott JJ, Ackerman MJ. An international compendium of mutations in the SCN5A-encoded cardiac sodium channel in patients referred for Brugada syndrome genetic testing. Heart Rhythm. 2010 Jan;7(1):33-46. doi: 10.1016/j.hrthm.2009.09.069. Epub 2009 Oct 8.).” The degree of certainty can also be expressed by the coverage, which was the following: 72.2 for GnomAD exomes coverage, and 34.4 for GnomAD genomes coverage. This is also included in the discussion.

Point 4: Could this truncated protein be active? Could it be expressed as an integral membrane protein? Could the expression of this truncated protein affect the function of the non-mutated channel?
Structure-function protein modeling should have been done or at least some mechanistic clues should be provided in the discussion.

Response 4: Although it is beyond the scope of this study to perform molecular/functional studies to show the precise effect of the truncating mutation, we have now added in the discussion the effect of other truncating variants found in the SCN5A gene. We now write: “Several other studies have reported pathogenic effects of premature stop codons in the SCN5A gene, leading to reduced protein expression, a non-functional protein that was confined in the cytosol rather than reaching the plasma membrane, and a complete loss of current (Maury et al., 2013) (Herfst et al., 2003) (Tfelt-Hansen et al., 2009).” We believe that this sentence could provide the reader with objective evidence as to what the function (or dysfunction) of the within described variant may be, based upon the function of very similar variants.

Reviewer 2 Report

This study reports a mutation in SCN5A that segregates with a phenotype of Brugada syndrome in a family. The authors have performed in silico analysis to suggest pathogenicity for the variant. The report is well-written and comprehensive; it could however benefit from certain additions so it does not just report yet another mutation in the sodium channel gene.

Major:

The assessment of family members section would benefit from the addition of a family pedigree.

Given how this is a case report – it is beyond the scope of this study to perform molecular/functional studies to show the precise effect of the truncating mutation. However, the authors should speculate on the mode of action of the mutation and of how loss of the C-terminus could affect protein interactions and otherwise the function of the channel. This addition would escalate the study from identification of yet another mutation in SCN5A to possible pathogenicity of it.

Minor:

Introduction: Given the phenotypic overlap between Brugada syndrome and arrhythmogenic cardiomyopathy and the very limited data implicating mutations in SCN5A in ARVC, I would be very hesitant to claim that Nav1.5 variants cause ARVC. It is more likely that the family reported in the literature in fact had BrS. I suggest that the authors either remove ARVC from the list of diseases or note that there is a single report suggesting this link without molecular/functional studies to confirm it.

Introduction: it might be important to mention that one of the down-sides in accurately diagnosing BrS in surviving family members is the fact that ajmaline testing can lead to false positives.

Assessment of family members: ‘her brother performed genetic testing’ – please rephrase to ‘her brother was subjected to genetic testing’ or ‘genetic testing was performed on the brother’

Has the proband had an MRI study? Was the arrhythmogenic substrate associated with the presence of fibrosis in the myocardium?

Author Response

Point 1: The assessment of family members section would benefit from the addition of a family pedigree.

Response 1: The family pedigree was shown in Figure 4.

Point 2: Given how this is a case report – it is beyond the scope of this study to perform molecular/functional studies to show the precise effect of the truncating mutation. However, the authors should speculate on the mode of action of the mutation and of how loss of the C-terminus could affect protein interactions and otherwise the function of the channel. This addition would escalate the study from identification of yet another mutation in SCN5A to possible pathogenicity of it.

Response 2: Although, as stated by the Reviewer, it is beyond the scope of this study to perform molecular/functional studies to show the precise effect of the truncating mutation, we have now added in the discussion the effect of other truncating variants found in the SCN5A gene, now writing, “Several other studies have reported pathogenic effects of premature stop codons in the SCN5A gene, leading to reduced protein expression, a non-functional protein that was confined in the cytosol rather than reaching the plasma membrane, and a complete loss of current (Maury et al., 2013) (Herfst et al., 2003) (Tfelt-Hansen et al., 2009).” We believe that this could provide the reader with objective evidence as to what the function (or dysfunction) of the within described variant may be, based upon the function of similar variants.

Point 3: Introduction: Given the phenotypic overlap between Brugada syndrome and arrhythmogenic cardiomyopathy and the very limited data implicating mutations in SCN5A in ARVC, I would be very hesitant to claim that Nav1.5 variants cause ARVC. It is more likely that the family reported in the literature in fact had BrS. I suggest that the authors either remove ARVC from the list of diseases or note that there is a single report suggesting this link without molecular/functional studies to confirm it.

Response 3: We have removed ARVC from the list of diseases.

Point 4: Introduction: it might be important to mention that one of the down-sides in accurately diagnosing BrS in surviving family members is the fact that ajmaline testing can lead to false positives.
Response 4: We now include in the introduction, “There have even been reports of false-positive ajmaline testing, making genetic diagnosis instrumental in determining whether these patients actually have the disease (J Brugada, P Brugada, R Brugada, The ajmaline challenge in Brugada syndrome: A useful tool or misleading information?, European Heart Journal, Volume 24, Issue 12, 1 June 2003, Pages 1085–1086, https://doi.org/10.1016/S0195-668X (03)00232-X).” Point 5: Assessment of family members: ‘her brother performed genetic testing’ – please rephrase to ‘her brother was subjected to genetic testing’ or ‘genetic testing was performed on the brother’

Response 5: We have changed this to read, “Her brother was subjected to genetic testing and a flecainide challenge elsewhere, both of which were negative.”

Point 6: Has the proband had an MRI study? Was the arrhythmogenic substrate associated with the presence of fibrosis in the myocardium?

Response 6: The proband was not subjected to an MRI study. However, he was subjected to an echocardiographic examination, which was normal. In the case presentation section, we now write, “Echocardiographic examination demonstrated normal morphological and functional parameters.”

Round 2

Reviewer 1 Report

The authors have adequately addressed all my comments.

Author Response

Thank you for your time to review our manuscript. We really appreciate your feedback. Best regards, The Authors